# Effects of Microelements on Downy Mildew (*Peronospora belbahrii*) of Sweet Basil

**DOI:** 10.3390/plants10091793

**Published:** 2021-08-28

**Authors:** Yigal Elad, Ziv Nisan, Ziv Kleinman, Dalia Rav-David, Uri Yermiyahu

**Affiliations:** 1Department Plant Pathology and Weed Research, Agricultural Research Organization, The Volcani Institute, 68 Hamakabim Rd, Rishon LeZion 7534509, Israel; zivnisan@gmail.com (Z.N.); dalia@volcani.agri.gov.il (D.R.-D.); 2Bikat HaYarden Research and Development, Tzevi Research Station, Bikat HaYarden 91906, Israel; ziv.kleinman@mail.huji.ac.il; 3The Robert H. Smith Faculty of Agriculture, Food and Environment, The Hebrew University of Jerusalem, Rehovot 76100, Israel; 4Gilat Research Center, Agricultural Research Organization, Volcani Institute, M.P. Negev, Gilat 8528000, Israel; uri4@volcani.agri.gov.il

**Keywords:** agrotechnical control, cultural control, integrated management, downy mildew, magnesium, manganese, microelements, plant disease, *Ocimum basilicum*, zinc

## Abstract

We recently demonstrated that spraying or irrigating with Ca, Mg and K reduces the severity of sweet basil downy mildew (SBDM). Here, the effects of Mn, Zn, Cu and Fe on SBDM were tested in potted plants. The effects of Mn and Zn were also tested under semi-commercial and commercial-like field conditions. Spray applications of a mixture of EDTA-chelated microelements (i.e., Fe-EDTA, Mn-EDTA, Zn-EDTA, Cu-EDTA and Mo) reduces SBDM severity. The application of EDTA chelates of individual microelements (i.e., Fe-EDTA, Mn-EDTA and Zn-EDTA) significantly reduces SBDM in potted plants. Foliar applications of Mn-EDTA and Zn-EDTA are found to be effective under semi-commercial conditions and were, thus, further tested under commercial-like conditions. Under commercial-like conditions, foliar-applied Mn-EDTA and Zn-EDTA decreased SBDM severity by 46–71%. When applied through the irrigation solution, those two microelements reduce SBDM by more than 50%. Combining Mg with Mn-EDTA and Zn-EDTA in the irrigation solution does not provide any additional disease reduction. In the commercial-like field experiment, the microelement-mixture treatment, applied as a spray or via the irrigation solution, was combined with fungicides spray treatments. This combination provides synergistic disease control. The mode of action in this plant–pathogen system may involve features of altered host resistance.

## 1. Introduction

Microelements are essential for plant development and are needed in lower concentrations than macro-elements. Microelements are active as co-factors of metalo-enzymes, and some of them play roles in the structural stability of membranes in vascular plants. Microelements are involved in metabolic processes, such as the synthesis of primary and secondary metabolites, energy metabolism, cell defense, the control of gene expression, hormone absorption and signal transduction [1]. Microelements affect the concentrations of phenols and lignin in plants [2] and also affect how plants respond to pathogens [3], including the induction of systemic acquired resistance. Substantial information on the effects of specific microelements on plant diseases was published in the second half of the 20th century. In the powdery mildew–cucumber patho-system, Mn and Cu were found to induce the release of Ca ions from membranes. Ca ions play a role in the defense system together with salicylic acid [4]. Interactions between microelements may affect both the pathogen and the host. For instance, Fe is essential for *Fusarium* pathogenicity, while Mn competes with the pathogen’s absorption of Fe [5]. Fe promotes the activity of a toxin of *F. oxysporum* var. *lycopersici*; whereas Cu inhibits the effect of Ca [6].

Mn is important for the development of plant resistance to foliar diseases [2]. It is involved in at least two steps in the biosynthesis of lignin, a process that is important for plant protection [7]. A single spray application of MnCl_2_ induced systemic-induced resistance to powdery mildew in cucumber plants and increased β-1,3-glucanase content even in non-infected plants [8]. Mn fertilization reduces the severity of foliar diseases such as downy mildews and powdery mildew [9].

Zn is important for the biosynthesis of proteins, energy production, transcription factor function and for maintaining the structure and integrity of membranes. More than 1200 proteins have been reported to contain, bind or translocate Zn [10,11]. There have been several reports on the effects of Zn on plant diseases. As an activator of Cu/Zn-SOD (superoxide dismutase), Zn is involved in protecting membranes against chilling damage by detoxifying superoxide radicals [12]. Protecting membranes against free radicals reduces the leakage of sugars and amino acids across membranes, which may limit parasite activity [2,12]. Zn reduces the susceptibility of cabbage (*Brassica oleracea*) and turnip (*Brassica rapa* subsp. *rapa*) to *Erysiphe polygoni* powdery mildew [13], as well as the severity of take-all disease in wheat, *Gaeumannomyces graminis* var. *tritici* [2].

Fe is involved in photosynthetic processes, mitochondrial respiration, assimilation of N; synthesis of hormones, such as jasmonic acid and gibberellin; the formation and elimination of reactive oxygen species (ROS) and resistance to plant pathogens [1]. Fe is also important for microorganisms, including microbial enzymes, that are important for plant infection [2,14]. Fe has been shown to reduce the severity of wheat rust and anthracnose in banana (*Musa acuminita*) [2]. Spray applications of Fe increased the resistance of apple (*Malus domestica*) and pear (*Pyrus communis*) to *Sphaeropsis malorum*, and the resistance of cabbage to *Olpidium brassicae* [5], but also increased the severity of take-all disease (*Gaeumannomyces graminis* var. *tritici*) in barley (*Hordeum vulgare*) [9].

Cu is essential for photosynthesis and mitochondrial respiration. It is also important for C and N metabolism, protection from oxidation damage, and cell-wall synthesis. Various biochemical reactions are dependent on Cu or Fe, and affected by their relative availability [15]. Enzymes such as polyphenol oxidase and phenolase, which are involved in lignin synthesis, require Cu, and Cu deficiency reduces the lignin content in plants [16]. Cu is a co-factor for the receptor of ethylene; thus, in cases of Cu deficiency, the affinity of the ethylene receptor to ethylene is drastically reduced [17]. Associations between low levels of Cu and increased disease severity have been reported for *Alternaria helianthi* infection of sunflower (*Helianthus annuus*), *Claviceps purpurea* infection (ergot disease) of barley and rye (*Secale cereale*) and *Septoria tritici* infection of wheat [5,18]. Increased Cu fertilization has been shown to reduce *Sclerotinia minor* infection in peanut (*Arachis hypogea*) [19] and *G. graminis* var. *tritici* infection in wheat [20].

We recently demonstrated that spraying or irrigating with N, Ca, Mg and K reduces the severity of sweet basil downy mildew (SBDM; [21,22]). SBDM is caused by *Peronospora belbahrii* [23]. It is a devastating disease in autumn and spring crops of sweet basil that has been reported to be affected by microclimate manipulation [24]. Chemical control options are limited and may fail due to the development of resistance [25]. One finding of that research was that SBDM is suppressed by concentrations of 3.0–4.94 mM Mg in the irrigation water under commercial-like conditions, as compared with a standard Mg concentration of 1.65 mM usually present in the irrigation water [21]. The purpose of the present study was to test the effects of the microelements Mn, Zn, Cu and Fe on SBDM in potted plants and under semi-commercial and field conditions. Selected microelements were also tested under commercial-like conditions, in combination with increased Mg concentration in the irrigation water and foliar applications of chemical fungicides. The research project was carried out in the same locations, in parallel and following the researches on effects of nitrogen and NH_4_^+^ fertilization [22] and of calcium, magnesium and potassium on sweet basil downy mildew (*Peronospora belbahrii*) [21]. These projects use similar methods and technical details.

## 2. Results

### 2.1. Effects of Foliar Applications of Microelements on Sweet Basil Downy Mildew (Experiments A1 and A2)

Two separate sets of potted-plant experiments were carried out to test the effects of foliar applications of a mixture of microelements and individual microelements on SBDM. A mixture of Fe-EDTA, Mn-EDTA, Zn-EDTA and Cu-EDTA (Koratin) reduced the severity of SBDM, when applied as a spray, at concentrations of 0.1–0.2%, but not at 0.4% (Figure 1a). The higher Koratin concentration was phytotoxic.

The individual EDTA chelates of Fe, Mn and Zn also provided significant disease control (Figure 1b). The control provided by the combination of microelements was not superior to that provided by the individual microelements (results not presented); thus, further work dealt with applications of individual microelements. The microelements, when applied as a mixture or individually, did not affect shoot yield.

### 2.2. Effects of Microelement Solutions on Sweet Basil Downy Mildew (Experiment C1, Autumn 2015–2016)

Experiments on the microelements’ effect on SBDM in location C were carried out in parallel to experiments in projects that dealt with nitrogen [22] and Ca, Mg and K effects on the disease [21]. In the present project, the severity of SBDM on untreated sweet basil plants reached 60% 151 days after planting. Spray applications of the mixture of EDTA-chelated microelements (Koratin, Fe-EDTA, Mn-EDTA, Zn-EDTA, Cu-EDTA and Mo, as (NH_4_)_2_MoO_4_) reduced disease severity until 129 days after planting, and foliar-applied fungicides reduced disease severity through the end of the experiment (Figure 2a). The calculation of AUDPC values (Figure 2b) revealed significant reductions in disease severity of 25.9%, 51.7% and 65.9% in the microelement, fungicides and combined microelement + fungicides treatments, respectively. The application of microelements did not significantly affect shoot yield.

A two-way statistical analysis of the results of the two main experiment components (i.e., fungicides sprays and microelement sprays) revealed a significant contribution of each of the main components to the reduction in SBDM severity, with the fungicides treatments providing a 52.7% reduction in severity, and the microelement treatments providing a 27.0% reduction in severity (*p* < 0.05). Calculation of the SF of the reduction in severity induced by the microelement and fungicides treatments revealed a significant synergistic effect for disease levels 147 days after planting (SF = 1.095) and for the calculated AUDPC (SF = 1.026).

We also examined the relationship between disease severity and the microelement concentrations in shoots sampled in individual plots in the experiment. At 94 days after planting, concentrations of Zn and Mn in the shoots were negatively correlated with SBDM severity (Figure 3a,b); whereas shoot concentrations of Cu and Fe were positively correlated with disease severity (Figure 3c,d). At 129 days after planting, Zn and Mn concentrations in the shoots were negatively correlated with SBDM severity, and the relationships between the concentrations of Cu and Fe and SBDM severity were insignificant (results not presented).

### 2.3. Foliar Application of Microelement Chelates to Sweet Basil Plants Grown in Containers (Semi-Commercial Experiments, Experiment B)

Sweet basil plants grown under semi-commercial conditions were treated with 0.1% Koratin. Disease in the untreated control reached an AUDPC of 153.6% × days, and the treatment reduced it by 18%. Further research concentrated on the effects of Mn and Zn on SBDM, because the effects of those microelements were consistent in the earlier potted-plant experiments (Site A) and since, under commercial-like conditions, the concentrations of Mn and Zn in sweet basil shoots were negatively correlated with SBDM severity. Therefore, spray applications of Zn-EDTA and Mn-EDTA were carried out over two months in plants grown in containers under semi-commercial conditions. Both treatments suppressed the severity of downy mildew on the peak disease sampling day, and at the end of the experiment (Figure 4a,b). The spray treatment that included both Mn and Zn did not affect shoot yield. Following the suppression of SBDM by Mn and Zn under semi-commercial conditions, we applied the two microelements in commercial-like experiments, as described below for experiments C2 through C5.

### 2.4. Foliar Microelement Treatments under Commercial-like Conditions (Experiment C2, Spring 2016)

In Experiment C2, the effects of foliar-applied Zn, Mn and fungicides on SBDM were compared under commercial-like conditions. The concentrations of Zn and Mn in the shoots were measured in untreated and treated plots at 69 days after planting. The shoot Zn concentrations in the untreated and treated plots were 37.48 ± 1.14 and 42.1 ± 2.1 mg/kg dry weight, respectively. The shoot Mn concentrations in the untreated and treated plots were 50.4 ± 1.84 and 68.5 ± 2.23 mg/kg dry weight, respectively. For each of the microelements, the difference between the treatments was significant (*p* ≤ 0.05). The fungicides treatment did not affect the concentrations of Mn and Zn in the shoots.

Disease severity 104 days after planting was 35 ± 4.1% in the untreated control and foliar applications of Zn, Mn and fungicides significantly (*p* ≤ 0.05) decreased disease severity by 59–71%. The calculated AUDPC was 773 ± 105.4 % x days in the untreated control and these treatments also significantly (*p* ≤ 0.05) reduced the calculated AUDPC by 46–71%.

In the untreated control, the total shoot yield was 6.22 ± 0.04 kg/m^2^ and the Grade A shoot yield was 3.88 ± 0.17 kg/m^2^. In the foliar-applied Zn plots, the total shoot yield was 6.70 ± 0.39 kg/m^2^ and the Grade A shoot yield was 4.25 ± 0.17 kg/m^2^. Finally, in the foliar-applied Mn plots, the total shoot yield was 7.23 ± 0.09 kg/m^2^ and the Grade A shoot yield was 4.39 ± 0.06 kg/m^2^. The yields of the spray treatments were significantly higher than that of the control (*p* ≤ 0.05).

### 2.5. Effects of Foliar and Fertigation Applications of Zn and Mn in Combination with Mg Supplied through Fertigation under Commercial-like Conditions (Experiment C3, Autumn 2016)

In a previous publication, we reported that SBDM could be suppressed by increasing the concentration of Mg in the irrigation solution, a treatment that was also associated with increased levels of Mg in the sweet basil shoots [21]. In the current experiment, we tested the effects of increased concentrations of Mn and Zn in the irrigation solution and foliar applications of Mn and Zn on SBDM when the irrigation solution contained either 1.65 mM Mg (basic level) or 4.94 mM Mg (supplemented).

Shoot concentrations of Mg in plants grown in the basic Mg irrigation treatments ranged between 0.83% and 1.02%, and for the plants grown in the supplemental Mg irrigation treatments, shoot Mg concentrations ranged between 1.20% and 1.29%. The concentration of Zn in the sweet basil shoots at 55 days after planting significantly increased as a result of the application of Zn through the irrigation solution, as observed in plants irrigated with both Mg concentrations, as well as plants treated with foliar-applied Mg that were irrigated with a solution that contained only the basic Mg concentration (Table 1). Interestingly, the increased concentration of Mg in the irrigation water increased the level of Zn in the shoots (Table 1, analysis not presented). A similar trend was observed for the Mn concentration in the shoots following the application of supplemental Mn through spray or fertigation treatments (Table 1), and as a result of the application of supplemental Mg (analysis not presented).

SBDM severity reached 52.5% at 93 days after planting. It was significantly reduced by Zn and Mn when supplied in the irrigation solution. The addition of these microelements to Mg-enriched water did not provide any further disease suppression (Figure 5a). Fungicides treatments drastically and significantly reduced SBDM severity to 7.5%, and only the fertigation-applied Zn supplement provided greater control (Figure 5b). Since the two-way ANOVA revealed a significant interaction between the microelements and the Mg treatment, each of the individual treatments is presented alone. Moreover, the combination of fertigation-applied microelements and fungicides treatments did not have any synergistic disease-control effect at 93 days after planting (results not presented).

The cumulative shoot yield for the control treatment reached 2.41 kg/m^2^ at 98 days after planting. None of the microelement-fertigation treatments resulted in any increases in yield. Yield in the fungicides treatment was 2.96 kg/m^2^, which was not significantly different from the yield of the plots that were not treated with any fungicide. The yields for the microelement + fungicides treatments were not significantly greater than the yields of plants treated only with the fungicides. The combination of 120 mg/L Mg and either Zn or Mn, provided an additive yield increase in both the treatments without fungicides and those with fungicide: 2.68 to 2.90 and 3.13 to 3.22 kg/m^2^, respectively.

Foliar-applied Zn significantly reduced SBDM severity (26.1%; Figure 6a). The fungicides treatments reduced the disease significantly while no additional suppression of the SBDM severity was achieved by the combination of fungicides with the foliar microelement applications (Figure 6b). Yield was not increased by the foliar application of microelements or fungicides (results not presented). Analysis of the microelement concentrations in the shoots of single replicates (experimental plots) did not reveal any relationship between microelement treatments and disease severity (data not shown).

### 2.6. Effects of Foliar Spray or Fertigation Applications of Zn and Mn Combined with Mg Supplementation under Commercial-like Conditions (Experiment C4, Spring 2017)

In the Spring of 2017 commercial-like experiment, we tested the effects of increased Mn and Zn in the irrigation solution on SBDM, when the irrigation solution contained either 1.65 mM (basic Mg level) or 4.94 mM (Mg supplemented). At 90 days after planting, the shoot Mg concentrations of plants grown in the basic Mg irrigation treatment ranged between 0.70% and 0.93%, and the shoot Mg concentrations for plants grown in the Mg-supplemented irrigation treatments ranged between 0.90% and 1.03%. The concentration of Zn in the sweet basil shoots increased significantly when Zn was applied through the irrigation solution, regardless of the level of Mg in that solution (Table 2). A similar trend was observed for the Mn concentration in the shoots following the application of Mn through the fertigation solution or by spray (Table 2).

The addition of Zn and Mn to the irrigation solution significantly suppressed SBDM severity (by more than 50%) at 104 days after planting, at both examined Mg levels (Figure 7a). The fungicides treatment significantly reduced disease and the combination of microelements and fungicide reduced disease significantly, as compared with the fungicides treatment alone (Figure 7a,b). There was no synergism between the increased Mg (4.94 mg/L) treatment and either of the microelement treatments in plots that were not treated with any fungicide. In the fungicide-treated plots, the combination of supplemental Mn and supplemental Mg had an additive effect (SF = 1.00). Synergism was obtained between the supplemental Zn treatment and the fungicides treatment (SF = 1.03). Differences in shoot yield were insignificant (results not presented).

Foliar applications of Zn and Mn significantly reduced SBDM severity by 44.0% to 48.0% (Figure 8a) and the fungicide significantly reduced disease severity by 45.6% (Figure 8b). The application of microelement spray treatments to fungicide-treated plants similarly reduced disease severity by 51.5%, as compared with the fungicides alone (Figure 8b). There was no synergism between the different foliar treatments. Cumulative yield in the untreated control was 4.558 kg/m^2^ at 110 days after planting. The yield was significantly increased (11.8%) by the application of Zn, and fungicides treatment increased yield by 7.8%.

The concentrations of microelements in the sampled shoots corresponded to the observed levels of disease severity. There were significant negative correlations between SBDM severity and the concentrations of Zn and Mn in the shoots (Figure 9).

### 2.7. Effects of Foliar Spray or Fertigation Applications of Zn, Mn and Zn Combined with Mn under Commercial-like Conditions (Experiment C5, Autumn 2017)

In the Autumn 2017 commercial-like experiments, we tested the effects of increased Mn, Zn and their combination in the irrigation solution (Figure 10a,b) and in spray (Figure 10c,d) on SBDM, when the plants were either not sprayed or additionally sprayed with fungicides. At 90 days after planting, the concentration of Zn in the sweet basil shoots increased significantly when Zn was applied through the irrigation solution and by spray (Table 3). Similarly, Mn concentration significantly increased in the shoots following the application of Mn through the fertigation solution or by spray (Table 3).

The addition of Zn and Mn to the irrigation solution significantly suppressed SBDM severity at 94 days after planting, but the combination of Zn with Mn did not result in additional SBDM reduction (Figure 10). The fungicides treatment significantly reduced disease and the combination of microelements and fungicide reduced disease significantly, as compared with the fungicides treatment alone (Figure 10).

Synergism was obtained between each of the supplemental -Zn, -Mn and –Zn + Mn treatments, and the fungicides treatment (Table 4). Differences in shoot yield were insignificant (results not presented).

The combined effects of the microelements and fungicides were estimated using the Abbott formula; SF = the synergy factor achieved by the combined treatment. When SF = 1, the interaction between the control measures is additive; when SF < 1, the interaction is antagonistic and when SF > 1, the interaction is synergistic.

## 3. Discussion

We recently demonstrated that the addition of Ca and Mg to the irrigation solution can reduce the severity of SBDM. We also demonstrated that although increased K concentration in the fertigation solution increased SBDM severity, foliar applications of KCl and K_2_SO_4_ suppressed SBDM [21]. In the present study, the effects of the microelements Mn, Zn, Cu and Fe (individually and in combination) on SBDM were tested. We found that even under commercial-like conditions, Zn and Mn provided a consistent, effective control of SBDM. The microelements that were applied at very low concentrations (i.e., 0.006–0.014% in the foliar treatments, and 1–2 mg/L in the irrigation solution) provided effective disease control.

There are no examples of previous research regarding the effects of microelements on SBDM or downy mildew of other crops. However, the effects of microelements have been examined in other pathosystems. For instance, foliar applications of B, Mn and Zn were tested for the control of tan spot disease (*Drechslera tritici-repentis*) in field-grown winter durum wheat. Following the foliar application of these microelements, the flag leaves of the treated plants had significantly fewer tan spots than those of the untreated plants. In general, there were minor differences between the effects of the Mn and Zn treatments. Similar to the present study, the treatments did not significantly affect wheat yield components [26]. Research with coffee rust (*Hemileia vastatrix*) yielded similar results; B, Zn, and Mn supplied to coffee plant (*Coffea* sp.) seedlings, reduced the severity of coffee rust on the leaves [27]. Severity of potato early blight (*Alternaria grandis*) was decreased in the potato (*Solanum tuberosum*) canopy when plants were treated with Zn and B + Zn. The authors of that work suggested that Zn plays a critical role in potato tolerance to early blight, and should be considered as a method for the control of potato early blight [28].

Increased levels of iron sulfate have been shown to decrease the severity of take-all, wheat crown and root rot caused by the soilborne fungus *Gaeumannomyces graminis,* and to enhance the biomass of the aerial parts of infected wheat plants. The effects of Zn and Cu fertilizers on the suppression of disease in wheat were weaker than the effects observed for Fe [29]; thus, our results are only partially similar to those of the wheat take-all research. Interestingly, wheat plants grown from seeds with a higher Mn content in soils that were naturally infested with *G. graminis*, were generally more vigorous and had an average of 11% less take-all [30]. Earlier, Brennan [31] performed five field experiments using five different levels of MnSO_4_ fertilizer, and found that take-all severity was decreased at two trial sites. In plots in which Mn levels were deficient, the application of Mn lowered the severity of take-all, had no effect on the incidence of take-all disease and increased the dry-matter and grain yields of the wheat plants. No beneficial effects were observed when Mn was added to soils that already contained adequate levels of Mn.

In another study involving bread wheat and durum wheat genotypes and another soilborne pathogen, *Fusarium solani*, the application of Zn had a positive effect on the plants’ resistance to *F. solani* root rot. Those researchers not only suggested that Zn nutrition can increase resistance to *F. solani*, but also that Zn deficiency should be avoided, in order to prevent susceptibility [32]. In contrast, a high foliar concentration of Zn, achieved through fertigation, increased the susceptibility of rice (*Oryza sativa*) to rice brown spot (*Bipolaris oryzae*) [33]. Nevertheless, in upland rice, low severity levels of panicle blast (*Pyricularia grisea*) were associated with higher concentrations of Zn and K in the plant tissue; no correlations were observed between the concentrations of other micronutrients in plant tissues and panicle blast severity [34].

In other research, cucumber (*Cucumis sativus*) plants were sprayed with MnSO_4_ four and six days before they were inoculated with *Podosphaera fuliginea* (the causal agent of powdery mildew) [35] and cucumber plants in a hydroponic system were sprayed four days before inoculation with the same pathogen [36]. In both of those studies, the Mn salt reduced the severity of cucumber powdery mildew. In a later study, foliar applications of MnSO_4_ to cucumber plants either before or after inoculation with *Colletotrichum lagenarium*, the causal pathogen of cucumber anthracnose disease, also suppressed the fungal infection of leaves and cotyledons. In that study, better disease control was observed when the treatment was applied three days prior to inoculation. All applied concentrations of Mn reduced disease severity to a similar degree [37].

The integrated effect of Zn applied with other nutrients on early blight (*Alternaria solani*) of tomato (*Solanum lycopersicum*) was also examined. In this patho-system, Zn applied in combination with Mg sometimes had no beneficial effect, and other times had a synergistic disease-control effect. Zn applied in combination with N, P and K had a significant synergistic effect on early blight [38]. In the present research, however, combining Mg with Zn and Mn in the irrigation solution did not provide any further reduction in disease severity. This is similar to the findings of our recent study, in which the combination of Ca and Mg did not provide improved SBDM control [21]. Moreover, in the present research, the combination of Zn and Mn did not provide better control than that provided by each microelement alone, regardless of whether the microelements were applied through the irrigation solution or as a foliar spray (Experiment C5). The combination of the tested microelements had a beneficial effect in only some of the commercial-like experiments; the disease control provided by Zn was synergistic with the fungicides treatment in Experiments C4 and C5. The control provided by Mn was synergistic with the fungicides treatment in Experiment C5. Therefore, a general conclusion regarding the benefit of this combination cannot be stated.

The application of Zn in combination with N, P and K for the potential control of early blight (*A. solani*) of tomato was associated with an increase in the total phenolic content and phenylalanine ammonia lyase (PAL) activity in plants. This suggests that these nutrients may have an additive effect on the production of salicylic acid, which may help to induce systemic resistance against a pathogen attack [38]. Similarly, in the coffee rust patho-system mentioned above, B, Zn and Mn significantly affected the concentration of total soluble phenols, while only Mn influenced the concentration of lignin [27]. The research that revealed that the foliar application of Mn could reduce the severity of cucumber powdery mildew [35,36] also involved a mechanistic study. A pre-inoculation foliar application of MnSO_4_ increased phenol oxidase activity and polyphenol oxidase activity in the treated leaves [35]. Moreover, the increased lignin, cellulose and pectin contents of the cell walls improved the leaf water status, which led the authors of that work to conclude that Mn nutrition could control cucumber powdery mildew by reinforcing the cell-wall structure and reducing the loss of water from infected leaves [36]. Similarly, Mn treatment of cucumber plants in the absence of a pathogen promoted lignification and the accumulation of ROS. A pre-inoculation Mn treatment of cucumber plants infected with *Colletotrichum lagenarium* enhanced pathogen-induced lignification, callose or ROS production, and reduced pathogen-induced cell death [37], indicating a role for Mn in induced-resistance mechanisms.

Foliar-applied Mn ameliorated the negative effects of orange rust (*Puccinia kuehnii*) on sugarcane (*Saccharum officinarum*). This effect was accompanied by increased sugarcane biomass production, and, in that system, Mn treatment caused direct damage to the fungal spores and improved lignin deposition in the mesophyll of the plants. During pathogenesis, Mn-sprayed leaves exhibited lower levels of oxidative stress, in addition to an improved structural organization of xylem and phloem vessels, as compared to the untreated control [39]. Since induced resistance may have a role in disease suppression by nutritional elements, further research should be directed towards this mode of action in various plant species.

## 4. Materials and Methods

### 4.1. Plants, Pathogen and Growing Conditions

Sweet basil (*Ocimum basilicum*) cv. Peri [40] was used for all of our experiments. The seedlings were grown in a commercial nursery (Shorashim, Mivtahim, Israel) and transplanted for the experiments 3 to 4 weeks after seeding in the nursery. The ‘Peri’ cultivar is known to be susceptible to *Peronospora belbahrii* [24]. Plugs of sweet basil seedlings were used, each containing three to five plants; hereafter, one plug is referred to as a “plant” as is common practice [41]. The experiments involving potted plants were performed in an experimental greenhouse at the Volcani Institute, Rishon LeZion, Israel (Site A). Experiments were also carried out using plants grown in containers under semi-commercial greenhouse conditions at the Volcani Institute, Rishon LeZion, Israel (Site B) and at the Tzvi R&D Experimental Station, Jordan Valley, Israel (Site C).

The sites of experimental work, the host plant and disease, plants maintenance, experimental designs, nature of data collected and statistical analysis were the same as described in earlier publications about parallel research projects [21,22]. Since experiments of the different research projects were carried out in parallel to each other in similar greenhouses at the same locations with similar planting dates, then technical details of the experiments were similar. In order to keep the research presented here independent, all technical issues are described below in detail as was earlier described for the parallel research works and described by Elad et al. [21,22].

At Site A, experiments were carried out in 2 L pots, each containing one plant. At Site B, experiments were carried out in polystyrene containers that were 0.8 × 0.8 × 0.17 m in size, with 12 plants in each container. Each container served as a replicate. At Site C, the experiments were carried out in 0.8 × 1.0 × 0.17 m containers; each experimental plot consisted of three containers and 24 plants were planted in each container (i.e., 72 plants per replicate). Sweet basil plants were planted in the pots and containers, which were filled with perlite (medium size, 1.2 mm, Agrifusia, Fertilizers & Chemicals Ltd., Haifa, Israel). For the potted-plant experiments involving foliar treatments, we used a potting mixture consisting of coconut fiber:tuff (unsorted to 8 mm; 7:3 vol.:vol.). The plants were irrigated to excess via a drip system two to four times a day, depending on the season, at a volume calibrated to lead to >30% water leaching. The daily irrigation volume was determined after analyzing the irrigation and drainage solutions once every 2 weeks, to prevent over-salinization or acidification of the root-zone solution. Plants in pots and containers were maintained according to the local extension service’s recommendations. All pot experiments were irrigated with fresh water (electrical conductivity (EC) < 1.0 dS/m). The tested elements were applied with the irrigation water (fertigation) or as a foliar spray, as described below and summarized in Table 5.

The oomycete inoculum originated from an experimental station in Rehov, Emek Hamaayanot, Israel [24] and was maintained on sweet basil plants. Conidia of *P. belbahrii* were collected in water by washing conidiating leaves of sweet basil plants that were kept in an experimental greenhouse at the Volcani Institute. The canopy of potted sweet basil plants was inoculated with a conidial suspension that contained 10^3^ cell ml^−1^ during the afternoon of the inoculation date. For symptom development, the plants were incubated at high RH (>95%) in the dark in a growth chamber at 22 ± 1 °C for 12 h, incubated in a greenhouse chamber at 22 ± 2 °C for 1 week, incubated at high RH (>95%) in the dark in a growth chamber at 22 ± 1 °C for 12 h and then incubated in a greenhouse chamber at 22 ± 2 °C. Potted sweet basil plants subjected to this artificial inoculation served as inoculum sources to ensure even inoculum loads across the greenhouse in the potted-plant experiments (Site A) and the semi-commercial container experiments (Site B). Disease severity (i.e., chlorosis, dry necrotic lesions and/or sporulation on the lower leaf side) was evaluated in the sweet basil canopy according to a 0–100 scale, in which 0 = no signs or symptoms and 100 = entire surface displays signs and/or symptoms, while, for instance, 20 and 50 are 20 and 50% canopy coverage by symptoms, respectively [21,22,24].

In the commercial-like experiments (Site C), downy mildew epidemics developed naturally during every experiment. Disease-severity evaluation in the field plots included all plants except for those along the edges of each plot. Disease severity was determined every 2 to 3 weeks in each plot of each experiment on a scale of 0 to 100, in which 0 = all plants visually healthy and 100 = all leaves on all plants in the plot showing typical downy mildew symptoms/signs of chlorosis, dry necrotic lesions and/or conidiating *P*. *belbahrii* on their lower side. Twenty and 50 levels of severity are 20 and 50% canopy coverage by symptoms, respectively [21,22,24].

Shoots that were >15 cm long were harvested and weighed. In the potted-plants experiment, yield was measured for each pot. In the semi-commercial and commercial-like experiments, shoots were harvested from each plot separately. The harvested shoots were sorted by quality and calculated Grade A yield per m^2^ figures are presented.

### 4.2. Foliar Applications of Microelement Chelates to Potted Sweet Basil Plants (Experiment A, Autumn and Spring 2014–2015)

The aim of these experiments was to study the effects of foliar applications of microelements on the development of SBDM in potted sweet basil plants. Treatments consisted of seven sweet basil plants that were grown in 2 L pots filled with potting mixture. The pot experiments were irrigated with fresh tap water (electrical conductivity (EC) < 1.0 dS/m). The plants were irrigated via a drip system three times a day at an excess rate (>30% leaching) using 2 L/h drippers. The daily irrigation volume was set after analysis of the irrigation and drainage solution once every 2 weeks to prevent over-salinization or acidification of the root-zone solution. All of the plants were fertigated with 5-3-8 fertilizer (N-P_2_O_5_-K_2_O; Fertilizers and Chemical Compounds Ltd., Haifa, Israel) at 90 g/m^3^ N. The water contained 0.7 mM Ca, 0.62 mM S and 0.95 mM Mg, as well as 0.023 mM B, 9.8 µM Fe, 4.9 µM Mn, 2.1 µM Zn, 0.31 µM Cu and 0.16 µM Mb (Fertilizers and Chemical Compounds Ltd.).

Foliar applications of microelements were made once a week with a 1 L hand sprayer. Plants were sprayed to runoff with the mixture of microelements 0.1–0.4% Koratin (Fertilizers and Chemical Compounds Ltd.). Koratin contains the following chelates: Cu-EDTA (0.2 g/L), Fe-EDTA (5.5 g/L), Zn-EDTA (1.35 g/L), Mn-EDTA (2.7 g/L) and Mo, as (NH_4_)_2_MoO_4_) (0.15 g/L). In another set of experiments, the chelates of the same microelements were sprayed individually, after one shoot harvest, at the following concentrations: Cu-EDTA (0.0095 mg/L), Fe-EDTA (0.03 mg/L), Zn-EDTA (0.006 mg/L) and Mn-EDTA (0.014 mg/L). Following two shoot harvests, when the plants in the experiments were 45 to 50 cm tall, they were artificially inoculated with *P. belbahrii* as described above.

### 4.3. Foliar Applications of Microelement Chelates to Sweet Basil Plants Grown in Containers (Commercial-like Experiment B, Autumn and Spring 2015–2016)

Sweet basil plants grown under semi-commercial conditions were treated with Zn-EDTA and Mn-EDTA over two months. The aim of these experiments was to study the effect of foliar applications of microelements on the development of downy mildew in sweet basil plants grown in polystyrene containers. The sweet basil plants were planted in 0.8 × 1.0 × 0.17 containers filled with perlite (medium size, 1.2 mm, Agrifusia, Fertilizers & Chemicals Ltd., Haifa, Israel), 12 plants (plugs) per container, in five replicates (one container = one replicate). The plants were irrigated with water, as mentioned above, via a drip-irrigation system three times a day at an excess rate (>30% leaching) using 2 L/h drippers. The daily irrigation volume was set after analysis of the irrigation and drainage solution once every 2 weeks to prevent over-salinization or acidification of the root-zone solution. All of the plants were fertigated with 5-3-8 fertilizer (N-P_2_O_5_-K_2_O) at 90 g/m^3^ N (Fertilizers and Chemical Compounds Ltd., Haifa, Israel). The water contained 0.7 mM Ca, 0.62 mM S and 0.95 mM Mg, as well as 0.03 mM B, 9.8 µM Fe, 4.9 µM Mn, 2.1 µM Zn, 0.31 µM Cu and 0.16 µM Mb (Fertilizers and Chemical Compounds Ltd.).

Foliar applications of microelements were conducted once a week with a backpack sprayer equipped with a conical nozzle. In one set of experiments, Koratin (0.1%) was sprayed to runoff. In a second set of experiments, Zn-EDTA (0.014 mg/L) and Mn-EDTA (0.006 mg/L) were applied (Fertilizers and Chemical Compounds Ltd., Haifa, Israel). Following two shoot harvests, when the plants in the experiments were 45 to 50 cm tall, they were artificially inoculated with *P. belbahrii* as described above.

### 4.4. Control of SBDM under Commercial-like Conditions (Experiments C)

At Site C, experiments were carried out in a polyethylene-covered greenhouse. Plants were irrigated daily according to local extension service recommendations. During the initial 5 days, plants were sprinkler-fertigated with 4.3 mM N (10% NH_4_^+^), 1.6 mM K and 0.65 mM P in the fertigation solution to aid their establishment. After that initial period, the plants were irrigated through drippers and fertilized with 8.57 mM N, 3.2 mM K and 0.65 mM P in water until the fertigation treatments were initiated, as described below. Throughout the growing season, the irrigation solution also contained 2 mM Ca, 1.65 mM Mg (unless supplemented to reach 4.94 mM Mg as mentioned below) and 0.09 mM B. Fertigation was performed from 1000 L tanks dedicated to each treatment, with a 17 mm drip-irrigation pipe that had a 2 L/h dripper embedded every 20 cm along its length. Spray treatments were made using a backpack sprayer equipped with a conical nozzle. Sprays were administrated until runoff once or twice a week. Experiments were carried out in the autumn (September–January) or spring (February–June) growing seasons. Each experiment was conducted in randomized blocks with four replicates. Each replicate consisted of three containers [21,22].

SBDM usually appeared 45–60 days after planting, reached a peak of severity and then began to become less severe. In the spring growing season, SBDM severity decreased to a low level due to the high temperatures and low humidity typical of the summer. In the autumn growing season, SBDM severity was low due to low temperatures [21,22]. The results from the greenhouse experiments refer to SBDM severity at a certain time after planting or to AUDPC over a certain period.

Chemical fungicides treatments included a rotation of two types of treatments. The first type of treatment was administered soon after harvest and consisted of a foliar application of a mixture of two fungicides: Canon (potassium phosphite 780 g/L, Luxembourg Industries (Pamol) Ltd. Tel Aviv, Israel applied at 0.3%) + Cabrio Duo (dimethomorph 72 g/L + pyraclostrobin 40 g/L) BASF, Ludwigshafen, Germany applied at 0.05%)) or, alternatively, Infinito ((fluopicolide 62.5 g/L + propamocarb-HCL 625 g/L), Bayer AG, Germany, applied at 0.1%). The second type of treatment was a foliar application of Canon at 1 week before harvest.

#### 4.4.1. Effects of Foliar-Applied Microelements under Field Conditions (Experiment C1, Autumn 2015–2016)

To examine the effects of the microelements on SBDM severity during the autumn–winter season, sweet basil was planted on 1 September 2015. The fertilizer mentioned above for fertigation with irrigation water was used in all treatments. Sweet basil plants were spray treated to runoff with the mixture of microelements 0.2% Koratin (Fertilizers and Chemical Compounds Ltd., Haifa, Israel) once a week. Koratin contains the following chelates: Fe-EDTA (5.5 g/L), Mn-EDTA (2.7 g/L), Zn-EDTA (1.35 g/L), Cu-EDTA (0.2 g/L) and Mo, as (NH_4_)_2_MoO_4_) (0.15 g/L).

Fungicides treatments were as described above. The EC of the fertigation solutions in the six treatments ranged from 1.91 to 2.45 dS/m and their pH values were 6.9 ± 0.1. Analysis of the nutritional elements was carried out at 94 days after planting (3 December 2013). Concentrations of the nutritional elements K, Ca, Mg and Cl in the shoots ranged between 3.34 and 4.90, 3.43 and 3.69, 0.42 and 0.52, and 0.66 and 0.82, respectively. Microelement concentrations in the shoots are presented in the Results (Figure 3). The sweet basil shoots were harvested on 14 October 2015, 1 November 2015 and 7 January 2016. Analysis of nutritional elements in sweet basil shoots was carried out on 3.12.15.

#### 4.4.2. Effects of Foliar-Applied Microelements under Commercial-like Conditions (Experiment C2, Spring 2016)

To test the effects of Zn and Mn on SBDM, sweet basil was planted on 23 February 2016. Fertigation started on 10 March 2016 with the basic fertilizer mentioned above and was the same in all treatments. Spray treatments with the microelement chelates Zn-EDTA (0.006%) and Mn-EDTA (0.014%) were applied weekly and fungicides were applied as described above; treatments were initiated on 24 April 2016.

Analysis of the nutritional elements was carried out at 69 days after planting (2 May 2016). The concentrations of N, P, K, Ca, Mg, Na and Cl in the shoots ranged between 4.06 and 4.31, 0.725 and 0.797, 4.75 and 5.53, 3.14 and 4.35, 0.80 and 0.85, 0.049 and 0.076, and 1.32 and 1.56%, respectively. The concentrations of Zn and Mn in the shoots are presented in the Results Section. The concentration of Fe in the shoots ranged between 110.5 and 117.6 mg/kg dry weight. The sweet basil shoots were harvested on 2 May, 19 May, 9 June, 5 July and 27 July.

#### 4.4.3. Effects of Zn and Mn Applied as Foliar Sprays or in the Irrigation Solution under Commercial-like Conditions (Experiment C3, Autumn 2016)

To test the effects of Zn and Mn on sweet basil downy mildew in an autumn crop, sweet basil was planted on 21 September 2016. Fertigation treatments were started on 28 September 2016. The basic fertilizer mentioned above was used in all treatments. The basic Mg concentration was 1.65 mM (40 mg/L) and an additional treatment of 4.94 mM (120 mg/L) Mg was made by adding MgCl_2_ to the irrigation solution [21]. The microelements were added to the irrigation solutions containing each of the Mg fertigation levels. Zn and Mn were added to the irrigation solutions at concentrations of 1 and 2 mg/L, respectively. Spray applications of 0.18% solutions of Zn and Mn were applied weekly to plots irrigated with a solution that contained 1.65 mM Mg. Fungicides treatments were as mentioned above and were applied in combination with the microelement fertigation treatments. The +/− fungicides treatments were located adjacent to one another. Fungicides were sprayed in the experimental plots starting from 16 November 2016. The sweet basil shoots were harvested on 26 October, 15 November and 19 December.

The EC of the fertigation solutions for the three Mg treatments ranged from 1.91 to 2.74 dS/m and their pH values ranged between 6.70 and 7.03. An analysis of nutritional elements in the shoots was carried out 55 days after planting (16 November 2016). The concentrations of N, P, K, Na and Cl in the shoots ranged between 4.57 and 5.67, 0.85 and 1.03, 4.13 and 4.70, 3.14 and 4.35, 0.07 and 0.08, and 0.77 and 1.47%, respectively. The concentrations of Zn and Mn in the shoots are presented in the Results Section (Table 1). The concentration of Fe in the shoots ranged between 121.1 and 151.6 mg/kg dry weight.

#### 4.4.4. Effects of Zn and Mn Applied under Field Conditions as Foliar Sprays or through the Irrigation Solution (Experiment C4, Spring 2017)

To test the effects of Zn and Mn on sweet basil downy mildew in the spring season, sweet basil was planted on 28 February 2017. Fertigation treatments started on 25 April 2017. The basic fertilizer described above was used in all treatments. The basic Mg concentration was 1.65 mM and an additional treatment of 4.94 mM was made by adding MgCl_2_ to the irrigation solution. The microelements added to the irrigation solution were 1 mg/L Zn-EDTA and 2 mg/L Mn-EDTA. Fungicides were sprayed in the plots from 30 May 2017 and Mn-EDTA (0.006%) and Zn-EDTA (0.014%) sprays were begun on 29 May 2017. The plots that received the +/− fungicides treatments were located adjacent to one another. Sweet basil shoots were harvested on 8 May, 29 May and 18 June.

The EC of the fertigation solutions for the two Mg treatments ranged between 1.85 and 2.87 dS/m and their pH values ranged from 6.95 to 7.12. An analysis of nutritional elements in the shoots was carried out at 90 days after planting (29 May 2017). The concentrations of N, P, K, Ca, Na and Cl in the shoots ranged between 4.68 and 4.95, 0.83 and 0.95, 4.90 and 6.02, 3.36, and 3.77, 0.07 and 0.08, and 1.15 and 2.03%, respectively. The concentrations of Fe and Cu in the shoots ranged between 95.6 and 106.7, and 10.7 and 12.6 mg/kg dry weight. The concentrations of Zn and Mn in the shoots are presented in the Results Section (Table 2).

#### 4.4.5. Effects of Zn and Mn Applied as Foliar Sprays or in the Irrigation Solution under Commercial-like Conditions (Experiment C5, Autumn 2017–2018)

To test the effects of Zn and Mn on sweet basil downy mildew in the spring season, sweet basil was planted on 26 September 2017. Fertigation treatments started on 25 October 2017. Fungicides sprays (as described above) were initiated on 9 November 2017 and microelements were applied individually or in combination (Mn-EDTA (0.006%) and Zn-EDTA (0.014%)), starting on 8 November 2017. The plots that received the +/− fungicides treatments were located adjacent to one another. The sweet basil shoots were harvested on 29 October 2017, 27 December 2017 and 24 January 2018.

The EC of the fertigation solutions were between 1.91 and 2.02 dS/m and their pH values ranged from 6.75 to 6.98. An analysis of the nutritional elements in the shoots was carried out at 92 days after planting (27 December 2017). The concentrations of N, P, K, Ca, Mg, Na and Cl in the shoots ranged between 3.97 and 4.73, 0.70 and 0.89, 4.83 and 6.05, 2.30 and 3.06, 0.47 and 0.80, 0.050 and 0.071, and 0.73 and 0.87%, respectively. The concentration of Fe and Cu in the shoots ranged between 87.4 and 106.1, and 10.4 and 11.9 mg/kg dry weight. The concentrations of Zn and Mn in the shoots are presented in the Results Section (Table 3).

### 4.5. Analysis of Elements

Shoots were sampled randomly in all experiments at harvest time from potted plants and from the commercial-like experiments for the determination of their mineral concentrations. The shoots were rinsed with distilled water and dried in an oven at 70 °C for 48 h. The dried plant material was ground and subjected to chemical analysis. The N and P concentrations in the shoots were determined after digestion with sulfuric acid and peroxide [42]. The concentrations of N and P were determined with an autoanalyzer (Lachat Instruments, Milwaukee, WI, USA). Concentrations of K, Na, Mg, Ca and microelements were analyzed with an atomic absorption spectrophotometer (Atomic Absorption PerkinElmer 460, Norwalk, CT, USA) after digestion with nitric acid and perchlorate spectrophotometry [43].

### 4.6. Data Analysis

The correlations between the concentration of a nutritional element and disease severity or between shoot concentrations of two selected nutritional elements were calculated using all individual pairs of data. Linear, exponential, logarithmic and polynomial correlations were examined. The formulas describing these correlations, correlation coefficient (*r*) values and α significance levels are noted in the Results Section.

Data in percentages were arcsine-transformed before further analysis. Area under the disease progress curve (AUDPC) values were calculated. Standard errors (SE) of means were calculated and are presented alongside with the number of degrees of freedom (df = *n* − 1 for controlled conditions experiments and df = *n* − 2 for correlations calculated for field conditions data). Disease-severity data and AUDPC data were analyzed by ANOVA and Tukey’s HSD test. Statistical analysis was performed using JMP 14.0 software (SAS Institute, Cary, NC, USA).

Disease reduction was calculated as follows:% disease reduction = 100 − 100 × (disease severity _T_ /disease severity _control_)
where T = the disease level in the treatment and control = the disease level in the untreated control.

The combined effect of the control measures used was estimated using the Abbott formula [44,45]. The expected disease reduction (control efficacy) and the combined suppressive activity were calculated as:CE_exp_ = a + b − a × b/100 and SF = CE_obs_/CE_exp_
where a = disease reduction due to one measure when applied alone; b = disease reduction due to the other measure when applied alone; CE_exp_ = expected control efficacy of the combined treatment, if the two measures act additively; CE_obs_ = observed disease reduction of the combined treatment; SF = the synergy factor achieved by the combined treatment. When SF = 1, the interaction between the control measures is additive; when SF < 1, the interaction is antagonistic and when SF > 1, the interaction is synergistic [44,45,46]. The same formula was used to calculate SF in the context of yield.

## 5. Conclusions

Zn and Mn, sprayed on plants or applied as part of the irrigation solution, contributed to the suppression of SBDM. The mode of action of this disease suppression merits further investigation. Similar to the further investigation of the suppression of SBDM by Mg, Ca and K [21], such future research should address possible host plant responses to the microelements and pathogen, including the upregulation of induced resistance pathways. The microelements are not toxic to the pathogen and the fact that only low concentrations of Zn and Mn are needed to reduce disease severity suggests that changes in plant host susceptibility may play a role in the observed effects.

## Figures and Tables

**Figure 1 plants-10-01793-f001:**
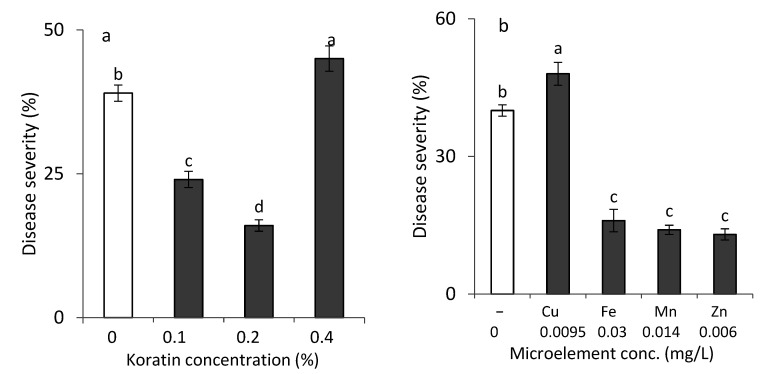
Effects of spray applications of solutions of EDTA chelates of microelements on sweet basil downy mildew (SBDM) severity when applied (**a**) in a mixture or (**b**) as single-microelement solutions. SBDM severity was evaluated on a 0–100% scale, in which 0 = healthy plants and 100% = plants completely covered by disease symptoms. Values followed by a different letter are significantly different according to one-way ANOVA and Tukey’s HSD test. Default significance levels were set at *p* ≤ 0.05. Bar = SE.

**Figure 2 plants-10-01793-f002:**
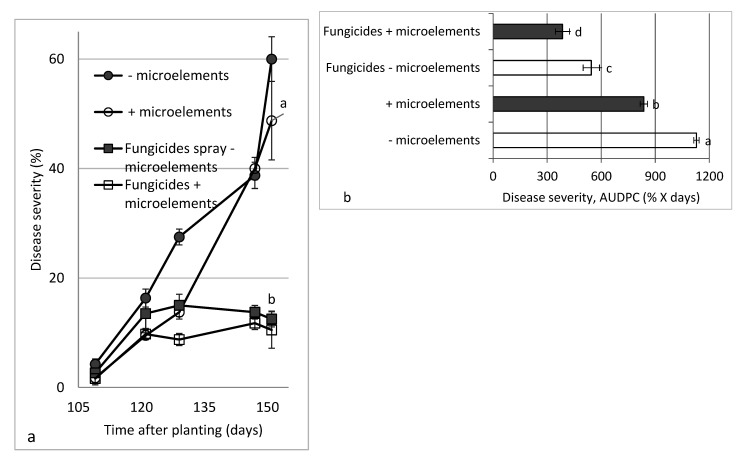
Effects of foliar applications of a microelement solution (Koratin: Fe-EDTA, Mn-EDTA, Zn-EDTA, Cu-EDTA and Mo mixture) and fungicides on sweet basil downy mildew (SBDM) severity in Experiment C1, Autumn 2015–2016. Treatments consisted of microelements (−/+ spray) coupled with fungicides treatments (−/+ spray). SBDM was evaluated (**a**) 109 to 151 days after planting and data are presented separately for (**b**) area under disease progress curve (AUDPC) through 42 days. Disease severity was evaluated on a 0–100% scale, in which 0 = healthy plants and 100% = plants completely covered by disease symptoms. Values in each graph and date followed by a different lower-case letter are significantly different according to one-way ANOVA with Tukey’s HSD test. Default significance levels were set at *p* ≤ 0.05. Bar = SE.

**Figure 3 plants-10-01793-f003:**
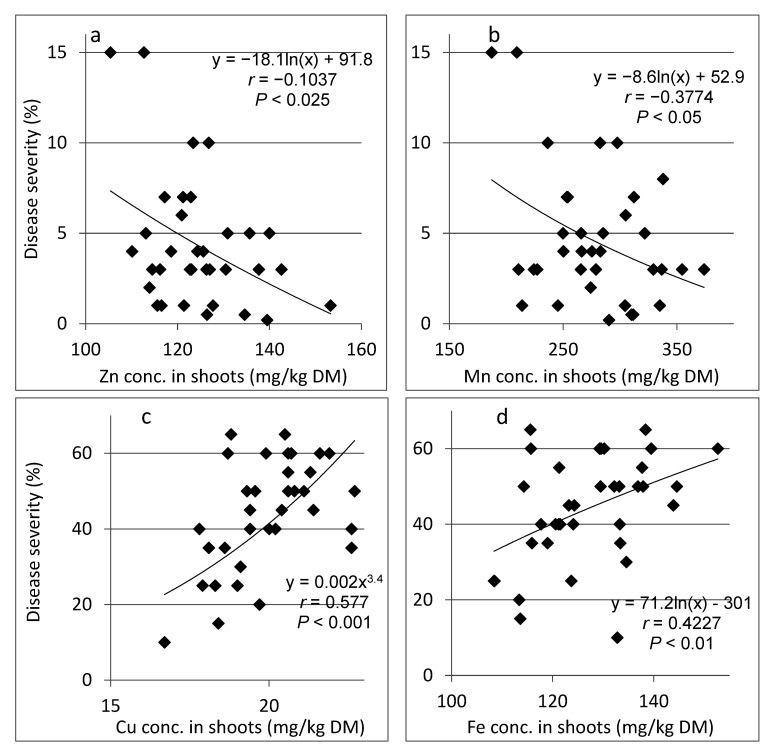
Relationships between the concentrations of microelements (**a**) Zn, (**b**) Mn, (**c**) Cu and (**d**) Fe in shoots of sweet basil sampled in single plots of Experiment C1 at 94 days after planting and the severity of sweet basil downy mildew (SBDM). SBDM severity was evaluated on a 0–100% scale, in which 0 = healthy plants and 100% = plants completely covered by disease symptoms. The best-fit regression formula is presented for each microelement. Pearson regression values (*r*) are presented along with significance levels (*p*).

**Figure 4 plants-10-01793-f004:**
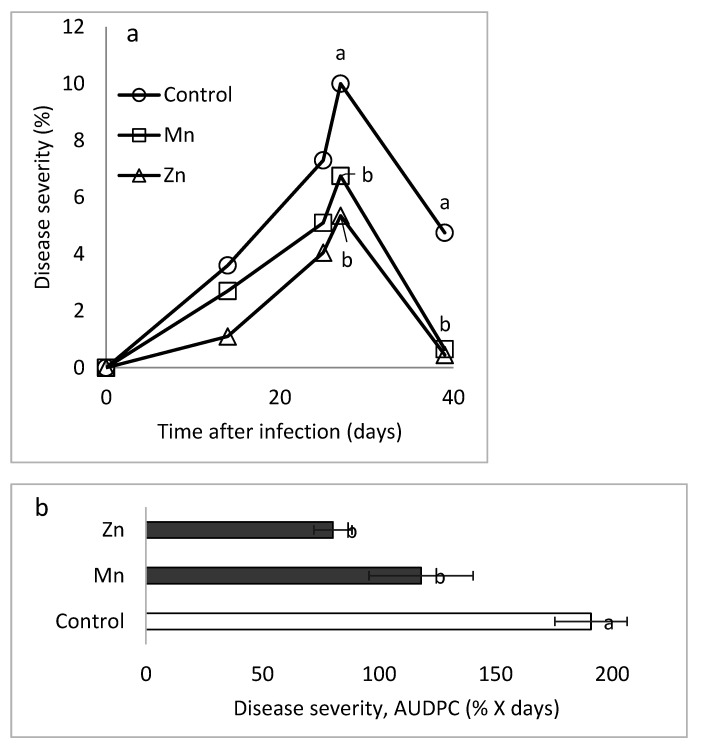
Effects of foliar applications of individual microelement chelates on the severity of sweet basil downy mildew (SBDM) under semi-commercial conditions. Zn-EDTA (0.006%) and Mn-EDTA (0.014%) were sprayed once a week over a two-month period. (**a**) Development of the disease and (**b**) area under disease severity curve (AUDPC) values through 39 days are presented. SBDM severity was evaluated on a 0–100% scale, in which 0 = healthy plants and 100% = plants completely covered by disease symptoms. Values followed by a different letter are significantly different according to one-way ANOVA with Tukey’s HSD test. Default significance levels were set at *p* ≤ 0.05. Bar = SE.

**Figure 5 plants-10-01793-f005:**
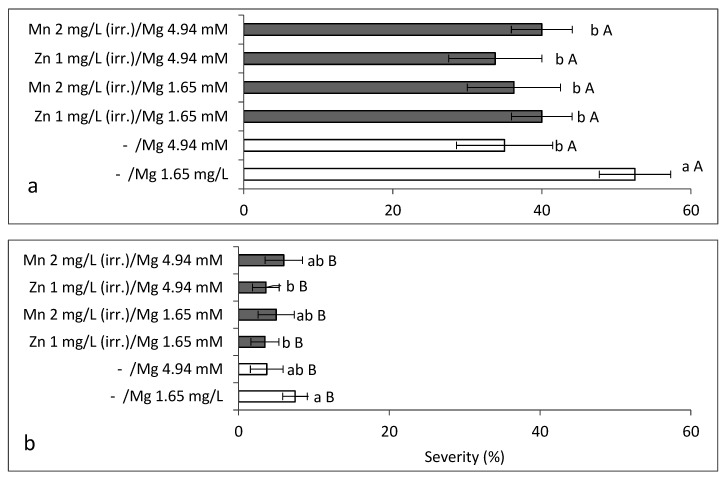
Effects of adding Zn-EDTA (1 mg/L; 1.64 mM) and Mn-EDTA (2 mg/L; 4.94 mM) to the irrigation solution on the severity of sweet basil downy mildew (SBDM) among plants grown under commercial-like conditions (Autumn 2016). Treatments were applied (**a**) alone or (**b**) together with fungicides. Severity of the SBDM at 93 days after planting was evaluated on a 0–100% scale, in which 0 = healthy plants and 100% = plants completely covered by disease symptoms. Values within each graph that are followed by a different lower-case letter and values followed by a capital letter in each pair of treatments without and with fungicides are significantly different according to two-way ANOVA with Tukey’s HSD test. The interaction between the main parameters (microelement application × Mg concentration) was significant, so detailed analyses of all of the combination treatments are presented. Default significance levels were set at *p* ≤ 0.05. Bar = SE.

**Figure 6 plants-10-01793-f006:**
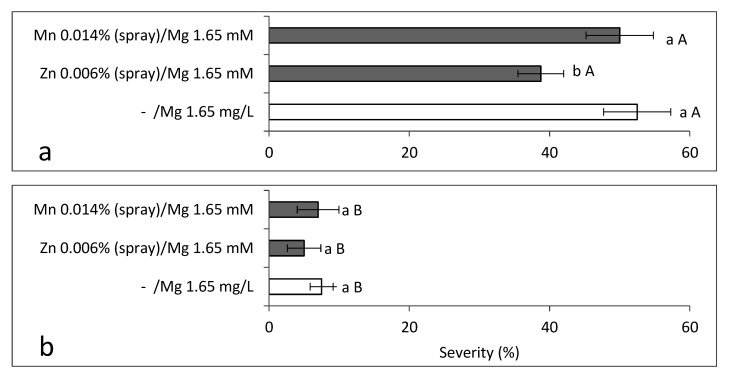
Effects of foliar applications of Zn-EDTA (0.006%), Mn-EDTA (0.014%) and fungicides on the severity of sweet basil downy mildew under commercial-like conditions (Autumn 2016). Plants were irrigated with a solution without added Mg (Mg concentration in water was regarded as 40 mg/L). Severity of SBDM at 93 days after planting (**a**) without fungicides and (**b**) with fungicides. Disease severity was evaluated on a 0–100% scale, in which 0 = healthy plants and 100% = plants completely covered by disease symptoms. Values within each graph that are followed by a different lower-case letter and values followed by a capital letter in each pair of treatments without and with fungicides are significantly different according to two-way ANOVA with Tukey’s HSD test. The interaction between the major parameters (fungicides × foliar-applied microelements) was significant, so detailed analyses of all of the combination treatments are presented. Default significance levels were set at *p* ≤ 0.05. Bar = SE.

**Figure 7 plants-10-01793-f007:**
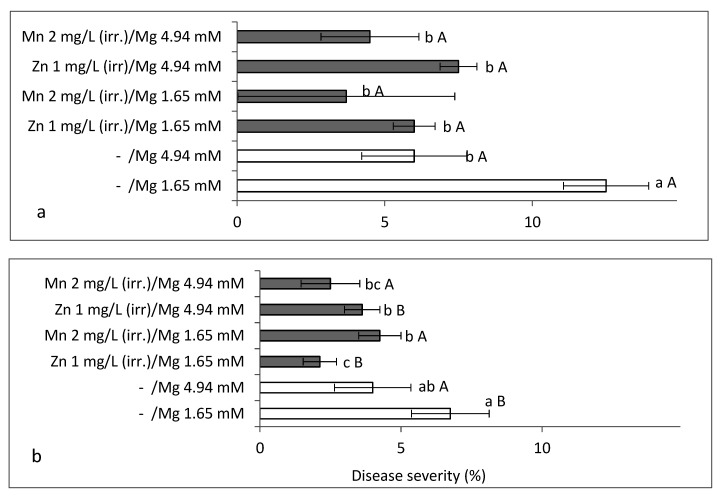
Effects of adding Zn-EDTA (1 mg/L) and Mn-EDTA (2 mg/L) to the irrigation solution so that it contained 1.65 and 4.94 mM/L of those microelements on sweet basil downy mildew (SBDM) severity under field conditions (Spring 2017). Treatments were applied (**a**) alone or (**b**) in combination with fungicides. SBDM severity was evaluated at 104 days after planting on a 0–100% scale, in which 0 = healthy plants and 100% = plants completely covered by disease symptoms. Values within each graph followed by a different lower-case letter and values followed by a capital letter in each pair of treatments without and with fungicides are significantly different according to two-way ANOVA with Tukey’s HSD test. Default significance levels were set at *p* ≤ 0.05. Bar = SE. DW—dry weight.

**Figure 8 plants-10-01793-f008:**
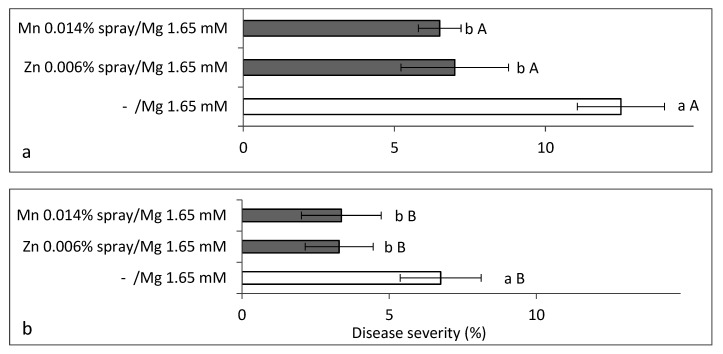
Effects of foliar applications of (**a**,**b**) Zn-EDTA (0.006%) and Mn-EDTA (0.014%) and (**b**) fungicides on the severity of sweet basil downy mildew (SBDM) under field conditions (Spring 2017). The plants were irrigated with no added Mg. (The Mg concentration in the irrigation solution was 1.65 mM.) Severity of SBDM at 104 days after planting (**a**) without fungicides and (**b**) with fungicides. Disease severity was evaluated on a 0–100% scale, in which 0 = healthy plants and 100% = plants completely covered by disease symptoms. Values within each graph that are followed by a different lower-case letter and values followed by a capital letter in each pair of treatments without and with fungicides are significantly different according to two-way ANOVA with Tukey’s HSD test. Default significance levels were set at *p* ≤ 0.05. Bar = SE.

**Figure 9 plants-10-01793-f009:**
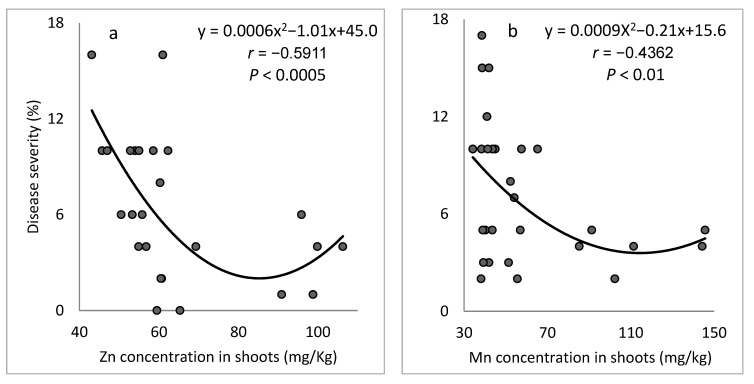
Relationship between the concentrations of microelements (**a**) Zn and (**b**) Mn in the shoots of sweet basil sampled in single plots of Experiment C4 and severity of sweet basil downy mildew (SBDM) at 104 days after planting. SBDM severity was evaluated on a 0–100% scale, in which 0 = healthy plants and 100% = plants completely covered by disease symptoms. The best-fit regression formula is presented for each microelement. The Pearson regression values (*r*) are presented along with significance levels (*p*).

**Figure 10 plants-10-01793-f010:**
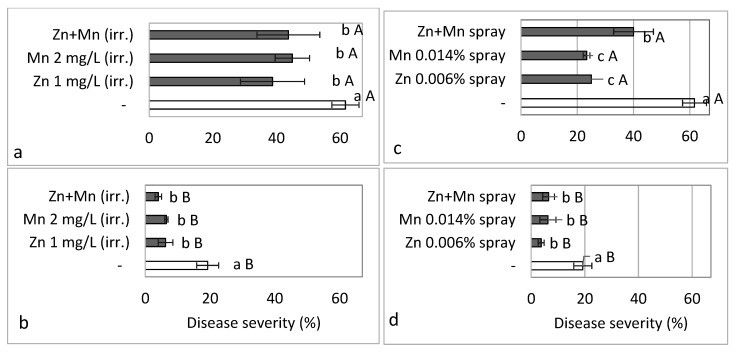
Effects of the addition of Zn-EDTA (1 mg/L) and Mn-EDTA (2 mg/L) to an irrigation solution containing 1.65 mM Mg (**a**,**b**) and spray of Zn-EDTA (0.006%) and Mn-EDTA (0.014%) (**c**,**d**) on the severity of sweet basil downy mildew (SBDM) under commercial-like conditions (Experiments C5, Autumn 2017–2018). The microelements were applied either alone or in combination; treatments were applied (**a**,**c**) without any fungicides or (**b**,**d**) with fungicides. SBDM severity at 94 days after planting was evaluated on a 0–100% scale, in which 0 = healthy plants and 100% = plants completely covered by disease symptoms. Values within each graph that are followed by a different lower-case letter and values followed by a capital letter in each pair of treatments without and with fungicides are significantly different according to one-way ANOVA with Tukey’s HSD test. Default significance levels were set at *p* ≤ 0.05. Bar = SE. DW—dry weight.

**Table 1 plants-10-01793-t001:** Concentrations of Zn and Mn in shoots of sweet basil plants that were grown under commercial-like conditions and treated with two different concentrations of Mg (applied through the irrigation solution), and Zn and Mn applied as a spray or through the irrigation solution (Experiment C3, Autumn 2016) at 55 days after planting.

Microelements in the Irrigation Solution		Zn in Shoots (mg/kg DW)		Mn in Shoots (mg/kg DW)	
Microelement	Concentration (mg/L)	Mg in Irrigation Solution (mM)			
		1.65	4.94	1.65	4.94
None	0	71.1 ± 2.78 c	78.8 ± 2.54 b	53.6 ± 4.41 b	49.0 ± 3.41 b
Zn	1	93.8 ± 1.73 b	118.2 ± 9.66 a	45.3 ± 4.05 b	63.8 ± 5.58 b
Mn	2	72.5 ± 2.49 c	77.6 ± 3.09 b	73.0 ± 9.95 a	115.3 ± 15.23 a
Foliar-applied microelements					
Microelement	Concentration (%)				
Zn	0.006	107.4 ± 7.05 a	Nm	49.9 ± 6.33 b	nm
Mn	0.014	69.5 ± 2.06 c	Nm	79.3 ± 5.04 a	nm

Values for each Mg treatment and each microelement treatment that are followed by a different letter are significantly different according to one-way ANOVA with Tukey’s HSD test. The default significance level was set at *p* ≤ 0.05. nm—not measured. DW—dry weight.

**Table 2 plants-10-01793-t002:** Concentrations of nutritional elements in the shoots of sweet basil plants grown under commercial conditions and treated with two Mg concentrations (applied through the irrigation solution) and with Zn and Mn (applied through the irrigation solution) at 90 days after planting (Experiment C4, Spring 2017).

Microelements in the Irrigation Solution		Zn in Shoots (mg/kg DW)		Mn in Shoots (mg/kg DW)	
Microelement	Concentration (mg/L)	Mg in Irrigation Solution (mM)			
		1.65	4.94	1.65	4.94
None	0	62.2 ± 1.11 b	66.0 ± 11.30 b	41.0 ± 1.56 b	44.0 ± 2.93 b
Zn	1	90.9 ± 12.86 a	96.1 ± 3.97 a	43.6 ± 4.09 b	50.0 ± 6.05 b
Mn	2	52.9 ± 2.13 c	55.9 ± 2.05 b	75.6 ± 11.11 a	123.2 ± 13.25 a

Values for each Mg treatment and each microelement treatment that are followed by a different letter are significantly different according to one-way ANOVA with Tukey’s HSD test.

**Table 3 plants-10-01793-t003:** Concentrations of nutritional elements in the shoots of sweet basil plants grown under commercial conditions and treated with Zn and Mn (applied through the irrigation solution) at 90 days after planting (Experiments C5, Autumn 2017, 2018).

Microelement			Concentration in Shoots (mg/kg DW)	
	In Irrigation Solution (mg/L)	Spray (%)	Zn	Mn
None	0	0	60.0 ± 3.90 c	71.4 ± 7.82 d
Zn	1		102.0 ± 6.03 a	70.5 ± 5.61 d
Mn	2		56.6 ± 4.65 c	94.2 ± 4.87 c
Zn + Mn	1 + 2		101.4 ± 11.73 a	102.1 ± 10.08 bc
Zn		0.006	76.0 ± 5.07 b	79.5 ± 4.37 d
Mn		0.014	59.5 ± 6.78 c	116.3 ± 14.06 ab
Zn + Mn		0.06 + 0.014	98.0 ± 4.77 a	124.8 ± 8.95 a

Values for each Mg treatment and each microelement treatment that are followed by a different letter are significantly different according to one-way ANOVA with Tukey’s HSD test.

**Table 4 plants-10-01793-t004:** Synergy factors (SF) of irrigation or spray treatments and foliar fungicides treatment at 94 days after planting.

Irrigation Supplement		Spray Treatment	
2 mg/L Mn	1.16	0.014% Mn	1.02
1 mg/L Zn	1.12	0.006% Zn	1.08
2 mg/L Zn + 1 mg/L Mn	1.20	0.006% Zn + 0.014% Mn	1.12

**Table 5 plants-10-01793-t005:** Experimental setup, factors tested, application methods and growing seasons ^1^.

Site	Expt. Code	Growing Setting	Microelements Tested	Additional Treatment	Application	Season
A	A1	Pots	Mixture (Koratin)		Spray	Autumn, Spring
A	A2	Pots	Fe, Cu, Zn, Mn		Spray	Autumn, Spring
B	B	Containers (semi-commercial)	Zn, Mn		Spray	Autumn, Spring
C	C1	Containers (commercial-like)	Mixture (Koratin)	+/− Fungicides	Spray	Autumn
C	C2	Containers (commercial-like)	Zn, Mn	Fungicides	Spray	Spring
C	C3	Containers (commercial-like)	Zn, Mn Zn, Mn Zn, Mn	Mg +/− Fungicides +/− Fungicides	Irrigation (Irr.) Irr., Spray Spray	Autumn
C	C4	Containers (commercial-like)	Zn, Mn Zn, Mn Zn, Mn	Mg +/− Fungicides +/− Fungicides	Irrigation Irr., Spray Spray	Spring
C	C5	Containers (commercial-like)	Zn, Mn, Zn+Mn Zn, Mn, Zn+Mn	+/− Fungicides +/− Fungicides	Irr., Spray Spray, Spray	Autumn

^1^ Experiments A were performed three times and B were performed twice with 5–7 replicates each time.

## Data Availability

The data that support the findings of this study are available from the corresponding author upon reasonable request.

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
