# Peer review of "Effects of Microelements on Downy Mildew (Peronospora belbahrii) of Sweet Basil"

_plants, 2021, doi:10.3390/plants10091793_

Round 1

Reviewer 1 Report

The authors of the manuscript “Effects of Microelements on Downy Mildew (Peronospora Belbahrii) of Sweet Basil” reported the suppression of SBDM following the application of microelements (Zn, Cu, Fe and Mn). They tested the microelements in potted plants and Mn/Zn also under semi-commercial and commercial-like field conditions. The treatments were applied as a spray or with the irrigation solution.

I have strong concerns about the scientific soundness, the methodology and the style.

Scientific soundness: The work focuses on evaluating the disease severity of SBDM following the application of microelements provided individually or in a mixture. The authors in some cases found a correlation between the amount of microelements in plant tissues and disease severity. However, they did not investigate the mechanisms underlying resistance. Much of the discussion talks about resistance mechanisms observed for the same microelements in other pathosystems but in this work there is no trace of experiments done to define resistance mechanisms. The work would seem to me more suitable for a technical/applied research journal.

Methodology: The disease index “0 = healthy plants and 100% = plants completely covered by disease symptoms” by which symptoms are measured does not seem defined in detail. I have doubts about the robustness of the data. Moreover, disease severity is measured at 69 or 93 or 94 or 104 or 129 or 147 days in the different experiments. This confuses the reader and adds concerns about the used methodology.

Style: Reading the manuscript is very difficult due to a redundancy that becomes extreme in materials and methods. Concentrations are expressed in a non-homogeneous way: %, mg/mL, ppm, mM. This confuses the reader and raises concerns about the comparison that was made in Figure 1.

Other concerns:

Figures 9 and 10 are included in the manuscript but are not mentioned or commented in the results. Experiment C5 is mentioned in Discussion but it is not mentioned in Results.

Line 217: Mg or Mn?

Controls are not expressed homogeneously in the figures. They are mentioned in different ways. Again, this confuses the reader.

It seems that the authors did not pay much attention in drafting the manuscript.

Author Response

English language and style

( ) Extensive editing of English language and style required
(x) Moderate English changes required
( ) English language and style are fine/minor spell check required
( ) I don't feel qualified to judge about the English language and style

Yes

Can be improved

Must be improved

Not applicable

Does the introduction provide sufficient background and include all relevant references?

(x)

( )

( )

( )

Is the research design appropriate?

( )

( )

(x)

( )

Are the methods adequately described?

( )

( )

(x)

( )

Are the results clearly presented?

( )

( )

(x)

( )

Are the conclusions supported by the results?

( )

( )

(x)

( )

Reviewer 1: Thanks for the remarks of reviewer 1. I found it useful and very assisting with improvement of the ms.  I related to all remarks and corrected the ms accordingly.

Reviewer 1: The authors of the manuscript “Effects of Microelements on Downy Mildew (Peronospora Belbahrii) of Sweet Basil” reported the suppression of SBDM following the application of microelements (Zn, Cu, Fe and Mn). They tested the microelements in potted plants and Mn/Zn also under semi-commercial and commercial-like field conditions. The treatments were applied as a spray or with the irrigation solution.

YE (Author): Thank you for the short summary. Additionally, the ms included combinations with fungicides and with Mg as well as Zn-Mn combination. The outcome of these treatments are important as well.

I have strong concerns about the scientific soundness, the methodology and the style.

Reviewer 1: Scientific soundness: The work focuses on evaluating the disease severity of SBDM following the application of microelements provided individually or in a mixture. The authors in some cases found a correlation between the amount of microelements in plant tissues and disease severity. However, they did not investigate the mechanisms underlying resistance. Much of the discussion talks about resistance mechanisms observed for the same microelements in other patho-systems but in this work there is no trace of experiments done to define resistance mechanisms. The work would seem to me more suitable for a technical/applied research journal.

YE (Author): In this manuscript and in earlier published works (Elad et al., 2021a; 2021b) with sweet basil downy mildew we already raised the possibility of induced resistance. As now mentioned in the discussion, we suggest the induced resistance as mode of action and as suggested by reviewer 2, this is also termed for future research. The fact that there is no additive disease suppression in combinations of nutritional elements point to a possibility for the mode of action as was described in our recent publication (Elad et al., 2021a) but it calls for specific mode of action work. Moreover, we already have some such mode of action results but these will go in a future manuscript once work will be finished. Nevertheless, I have an experience with many publications and I can assure the reviewer the high value of the current ms presented results. The ms contains very intensive experiments that were carried over four years and 5 growing seasons and each of the field experiments resulted in many data.

References:

Elad, Y.; Kleinman, Z.; Nisan, Z.; Rav-David, D.; Yermiyahu, U. Effects of calcium, magnesium and potassium on sweet basil downy mildew (Peronospora belbahrii). Agronomy 2021a, 11, 688. https://doi.org/10.3390/ agronomy11040688

Elad, Y.; Nisan, Z.; Kleinman, Z.; Rav-David, D.; Yermiyahu, U. The effect of nitrogen and NH4+ fertilization on Peronospora belbahrii downy mildew of sweet basil. Phytoparasitica 2021b, https://doi.org/10.1007/s12600-021-00922-y

Reviewer 1: Methodology: The disease index “0 = healthy plants and 100% = plants completely covered by disease symptoms” by which symptoms are measured does not seem defined in detail. I have doubts about the robustness of the data.

YE (Author): The scale of 0 to 100% is very detailed. Every 'degree' of percent count. The evaluation is carried out by experienced experts and the data is robust. See also many other publications of research work below (I cite only the ones from 2019). I corrected  the description of the M&M in order that it will be more clear and added one reference.

  1. Rav David, D., Yermiyahu, U., Fogel, M., Faingold, I. and Elad, Y. (2019) Plant nutrition for management of white mold in sweet basil. Phytoparasitica 47: 99-115 DOI: 10.1007/s12600-019-00716-3
  2. Oliva, M., Hatan, E., Kumar, V., Galsurker, O., Nissim-Levi, A., Ovadia, R., Galili, G., Lewinsohn, E., Elad, Y. and Alkan, N. (2019) Increased phenylalanine levels in plant leaves reduces susceptibility to Botrytis cinerea. Plant Science 290: 110289. https://doi.org/10.1016/j.plantsci.2019.110289
  3. Gupta, R., Leibman-Markus, M., Marash, I., Kovetz, N., Rav-David, D., Elad, Y. and Bar, M. (2021) Root zone warming represses foliar diseases in tomato by inducing systemic immunity. Plant, Cell & Environment. 44:2277–2289. https://doi.org/10.1111/pce.14006
  4. Elad, Y., Kleinman, Z., Nisan, Z., Rav-David, D. and Yermiyahu, U. (2021) Effects of calcium, magnesium and potassium on sweet basil downy mildew (Peronospora belbahrii). Agronomy 11, 688. https://doi.org/10.3390/ agronomy11040688.
  5. Elad, Y., Nisan, Z., Kleinman, Z., Rav-David, D. and Yermiyahu, U. (2021) The effect of nitrogen and NH4+ fertilization on Peronospora belbahrii downy mildew of sweet basil. Phytoparasitica DOI 10.1007/s12600-021-00922-y
  6. Omer, C., Nisan, Z., Rav-David, D. and Elad, Y. (2021). Effects of agronomic practices on the severity of sweet basil downy mildew (Peronospora belbahrii). Plants 2021, 10, 907. https://doi.org/10.3390/plants10050907 Suplementary Materials: https://www.mdpi.com/2223-7747/10/5/907#supplementary ; file:///C:/Users/elady/AppData/Local/Temp/Temp1_plants-10-00907-s001%20(4).zip/plants-1191251-supplementary.pdf

Reviewer 1: Moreover, disease severity is measured at 69 or 93 or 94 or 104 or 129 or 147 days in the different experiments. This confuses the reader and adds concerns about the used methodology.

YE (Author): Experiments under field conditions with natural disease infection, carried out in different years and different seasons have different rated of host plant development, harvest times, epidemics severity and control efficacy. The results of downy mildew levels were evaluated at best time for phenomena evaluation under restrictions of good phytopathological research work. It is not right to evaluate at the same date at all epidemics. To accommodate the so called confusion of evaluation dates, all treatments at a single experiment were evaluated at the same time by the same evaluators.

Reviewer 1: Style: Reading the manuscript is very difficult due to a redundancy that becomes extreme in materials and methods. Concentrations are expressed in a non-homogeneous way: %, mg/mL, ppm, mM. This confuses the reader and raises concerns about the comparison that was made in Figure 1.

YE (Author): These are the ways to describe concentrations in plant nutrition sciences. Added nutritional elements to irrigation water are usually described with M and mM units. Added chelates to irrigation water are described by mg/L.  Spray of nutritional salts (and fungicides) are described with % units. See also references in the manuscript and in the response to earlier remarks.

In order to avoid the confusion I replaced ppm with mg/L in the units describing concentrations of chelated microelements.

Reviewer 1: Other concerns:

Figures 9 and 10 are included in the manuscript but are not mentioned or commented in the results. Experiment C5 is mentioned in Discussion but it is not mentioned in Results.

YE (Author): Right, Thank you, the text was written and included in our last draft while somehow it disappeared from the submitted version. I am sorry for this technical mistake. The results section was corrected accordingly.

Reviewer 1: Line 217: Mg or Mn?

YE (Author): Mg, as written.

Reviewer 1: Controls are not expressed homogeneously in the figures. They are mentioned in different ways. Again, this confuses the reader.

YE (Author): Thank you, I agree, controls are now marked with empty bars as was done for the majority of the graphs in the previous version. I also corrected some graphs for other style matters and for the sake of uniformity.

Reviewer 1: It seems that the authors did not pay much attention in drafting the manuscript.

YE (Author): I am sure that the corrected manuscript is in good order.

As for the English language: The manuscript was edited by a native English editor before the original submission. Because of the remark of the reviewer the ms was corrected again by an native English scientist in our department and few corrections were made.

Reviewer 2 Report

This article is interesting, useful, and well prepared. The paper studies the effects of microelements on downy mildew (Peronospora belbahrii) of sweet basil.  The microelements are not toxic to the pathogen and the fact that only low concentrations of Zn and Mn are needed to reduce disease severity suggests that changes in plant host susceptibility may play a role in the observed effects. 

Introduction it is in line with the Instructions for the Authors. The methodology is also corresponding to the experimental part points of interest. Results are representative and comprehensive.

Discussion is also appropriate, but a more precise direction of future research should add more value.

References are in line with the scientific demonstration of the main issues.

It is an interesting research work and it is recommended to publish it as a first from a series with incoming findings.

Yours sincerely,

Author Response

Open Review

English language and style

( ) Extensive editing of English language and style required
( ) Moderate English changes required
( ) English language and style are fine/minor spell check required
(x) I don't feel qualified to judge about the English language and style

Yes

Can be improved

Must be improved

Not applicable

Does the introduction provide sufficient background and include all relevant references?

(x)

( )

( )

( )

Is the research design appropriate?

(x)

( )

( )

( )

Are the methods adequately described?

(x)

( )

( )

( )

Are the results clearly presented?

(x)

( )

( )

( )

Are the conclusions supported by the results?

(x)

( )

( )

( )

Reviewer 2 Comments and Suggestions for Authors – all marked and corrected accordingly:

Reviewer 2: This article is interesting, useful, and well prepared. The paper studies the effects of microelements on downy mildew (Peronospora belbahrii) of sweet basil.  The microelements are not toxic to the pathogen and the fact that only low concentrations of Zn and Mn are needed to reduce disease severity suggests that changes in plant host susceptibility may play a role in the observed effects. 

Introduction it is in line with the Instructions for the Authors. The methodology is also corresponding to the experimental part points of interest. Results are representative and comprehensive.

YE (Author): Thank you, no correction made.

Reviewer 2: Discussion is also appropriate, but a more precise direction of future research should add more value.

YE (Author): An emphasis on future, induced resistance research was made.

Reviewer 2: References are in line with the scientific demonstration of the main issues.

YE (Author): Thank you, no correction made.

Reviewer 2: It is an interesting research work and it is recommended to publish it as a first from a series with incoming findings.

YE (Author): Thank you, yet this is the third manuscript in line of nutrition effect on sweet basil downy mildew.

Reviewer 3 Report

Dear authors,

The authors of the manuscript entitled “Effects of Microelements on Downy Mildew (Peronospora Belbahrii) of Sweet Basil” reported the effect of mineral nutrition on disease and yield. The Manuscript provides an advancement in the field of research and shows that adoption of cultural practice can be helpful in sustainable management of P. belbahrii.

Some concerns need to be addressed:

The manuscript has too much information making them difficult to read. The results should be summarized highlighting important findings. I suggest including secondary results in the supplementary material. Authors should present only AACPC data, since AUDPC is a summary of disease intensity over time. For example, in figure 2, the graphs (a,b,c) show the same results in different formats. I suggest authors remove (figure 2 -b). It would be interesting to do the same for the other results. Remove (figure 4-b)

Figure 4 - a) add error bars on all data points

Line 117- What is the concentration of the mixture? Standardize the terms (mixture/Koratin) throughout the manuscript.

Lines 189-194. Put the text results in a table. Delete figure 5

As the combination of Mg with Mn and Zn in the irrigation did not provide any additional control, I recommend not showing the results in figures. Just mention them in the text.

Author Response

Open Review

English language and style

( ) Extensive editing of English language and style required
( ) Moderate English changes required
( ) English language and style are fine/minor spell check required
(x) I don't feel qualified to judge about the English language and style

Yes

Can be improved

Must be improved

Not applicable

Does the introduction provide sufficient background and include all relevant references?

(x)

( )

( )

( )

Is the research design appropriate?

(x)

( )

( )

( )

Are the methods adequately described?

(x)

( )

( )

( )

Are the results clearly presented?

( )

(x)

( )

( )

Are the conclusions supported by the results?

(x)

( )

( )

( )

Reviewer 3: Thank you, I corrected the ms along the suggestions of the review including omition of some graphs.

Reviewer 3 Comments and Suggestions for Authors alongside response of the authors.

Reviewer 3: Dear authors, The authors of the manuscript entitled “Effects of Microelements on Downy Mildew (Peronospora Belbahrii) of Sweet Basil” reported the effect of mineral nutrition on disease and yield. The Manuscript provides an advancement in the field of research and shows that adoption of cultural practice can be helpful in sustainable management of P. belbahrii.

Some concerns need to be addressed:

The manuscript has too much information making them difficult to read. The results should be summarized highlighting important findings. I suggest including secondary results in the supplementary material.

YE (Author): Thank you, Indeed there are many results. As suggested below by the reviewer, four graphs were omitted  but other than the suggested Figures we actually do not have more 'secondary results' as we presented a series of experiments that were conducted in time lines over years and growth seasons. Furthermore, each of the field experiments Figure represents a development of experimentation over the previous experiment and represent a whole season work.

Reviewer 3: Authors should present only AACPC data, since AUDPC is a summary of disease intensity over time.

YE (Author): I am not sure what AACPC data are. Nevertheless, AUDPC is a kind of summation of results that present in one value an entire epidemic development. Presenting all the data of such an epidemic will indeed result in a too large results section. AUDPC is a common way to shortly present an entire epidemic.

Reviewer 3: For example, in figure 2, the graphs (a,b,c) show the same results in different formats. I suggest authors remove (figure 2 -b). It would be interesting to do the same for the other results.

YE (Author): Indeed, Fig. 2 b data are also presented in Fig. 2 a. Fig 2b was removed and Fig. 2c was made the new Fig. 2b.

Reviewer 3: Remove (figure 4-b)

YE (Author): Indeed, Fig. 4 b data are also presented in Fig. 4 a. Fig 4b was removed and Fig. 4c was made the new Fig. 4b.

Reviewer 3: Figure 4 - a) add error bars on all data points.

YE (Author): The removal of the original Fig 4b resulted also in placing statistical analysis in Fig. 4a so there is no need for the additional SE.

Reviewer 3: Line 117- What is the concentration of the mixture? Standardize the terms (mixture/Koratin) throughout the manuscript.

YE (Author): Microelements mixture/Koratin was corrected and standardized throughout the ms.

Reviewer 3: Lines 189-194. Put the text results in a table. Delete figure 5.

YE (Author): Fig. 5 was omitted and the results are mentioned in the text only. In order to reduce manuscript volume we did not add a table,

Reviewer 3: As the combination of Mg with Mn and Zn in the irrigation did not provide any additional control, I recommend not showing the results in figures. Just mention them in the text.

YE (Author): I think that these results are very important and hope that it can remain in manuscript.

Round 2

Reviewer 1 Report

The manuscript is still hard to read because of the large amount of results not clearly presented, the redundancy of the information (especially in MM) and the inadequacy of Discussion that is focused on the mechanisms of resistance (the manuscript does not investigate the mechanisms of resistance).

Reviewer 3 Report

Dear authors,

The manuscript entitled "Effects of microelements on downy mildew (Peronospora belbahrii) of sweet basil" are appropriate for publication.